# Serious and Progressive Neuropathy Presumably Post-Shingrix Vaccination

**DOI:** 10.3390/reports7010005

**Published:** 2024-01-13

**Authors:** Michael J. Wons, Avani Vaghela, Amna Khalid, Benjamin D. Brooks

**Affiliations:** Department of Biomedical Sciences, Rocky Vista University, Ivins, UT 84738, USA; michael.wons@rvu.edu (M.J.W.); avani.vaghela@ut.rvu.edu (A.V.); amna.khalid@ut.rvu.edu (A.K.)

**Keywords:** Shingrix, neuropathy, vaccine, varicella

## Abstract

We present a case of serious and progressive neuropathy shortly following the administration of the herpes zoster subunit (HZ/su) vaccine, otherwise known as Shingrix. The progressive neuropathy occurred a week following the vaccination and progressed until discharge 16 days post-admittance. The patient’s mild symptoms persist. The development of neuropathy following HZ/su administration is exceedingly rare, with an attributable risk of three cases per million vaccines administered. A black box warning was issued for this indication, although diagnosis and treatment were not confirmed for this patient. Reporting cases like this is crucial for a comprehensive understanding of vaccine risks and to characterize the underlying etiology of these serious adverse events.

## 1. Introduction

Herpes zoster, commonly known as shingles, is a reactivation of the varicella zoster virus (VZV), which also causes chickenpox. The estimated lifetime incidence of herpes zoster is around 10 to 20 percent [1]. Mortality from herpes zoster is rare, with a reported incidence of 0 to 0.47 per 100,000 people per year [1]. Morbidity data from a general practitioner (GP) network in France showed that 1% of patients with HZ were hospitalized and the death rate was 0.2/100,000. The estimated annual medical cost to treat herpes zoster in the United States alone is USD 1.1 billion [2]. To reduce the morbidity and mortality associated with shingles, the Advisory Committee on Immunization Practices (ACIP) recommends that immunocompetent adults over the age of 50 be vaccinated against herpes zoster virus infections via the HZ/su vaccine, otherwise known as the Shingrix vaccine.

The Shingrix vaccine against herpes zoster (HZ) was approved by the FDA in 2017 for adults aged 50 years and older. It combines the varicella zoster virus glycoprotein E antigen with the AS01B adjuvant system. The shingles vaccine is administered as a two-dose series. For most people, the second dose should be administered 2 to 6 months after the first dose. Some people who have, or will have, a weakened immune system may receive the second dose 1 to 2 months after the first dose.

In 2021, the FDA issued a black box warning for Shingrix regarding the potential increased risk of Guillain–Barré syndrome (GBS) following Shingrix vaccination. GBS is a rare neurological disorder associated with viral infections and vaccination in which the body’s immune system damages nerve cells, causing muscle weakness and sometimes paralysis [3]. Analysis of Vaccine Adverse Event Reporting System (VAERS) data and clinical trial data showed an increased risk of GBS in the 6 weeks after vaccination, mostly in people aged 50 years and older. The estimated attributable risk is 3 excess GBS cases per million Shingrix vaccine doses administered to adults aged 65 years and older. Given the availability of a vaccine against HZ and the serious complications related to HZ disease, the benefits of Shingrix still outweigh the potential risks for protection against HZ and its complications in adults aged 50 years and older. However, it is important for patients and providers to be aware of the potential for GBS following vaccination, albeit rarely, to guide monitoring and prompt treatment if necessary. The continued reporting of adverse events is critical to define the safety profile of Shingrix fully.

Of the systemic adverse reactions experienced, patients most commonly describe myalgias, fatigue, and headaches, but may also experience localized pain at the injection site, erythema, and swelling [4]. There have also been reports of patients experiencing uveitis sarcoidosis post-vaccine administration [5]. Neuropathy is a recognized adverse event of the Shingrix vaccine [6]. The QS-21 component in the AS01 adjuvant in the Shingrix vaccine has been correlated with the neurological adverse events of the vaccine [7]. It is hypothesized that, through molecular mimicry from vaccine administration, patients experiencing adverse reactions may express immune cross-reactivity, leading to autoimmune-like symptoms [8]. In this paper, we present the case of a 64-year-old female patient who presented with a unique array of primarily neurological symptoms.

## 2. Detailed Case Description

A 64-year-old woman with a previous medical history significant for tobacco usage, hypertension, hypothyroidism, insomnia, post-menopausal bleeding, urinary calculus, and urinary retention presented with a series of symptoms following Shingrix vaccine administration eleven days post-vaccination. The initial presentation of symptoms included bilateral paresthesia through her wrists, arms, and lower extremities, as well as headaches, fevers, and muscle aches in her hips and back. The patient received no other medications or vaccinations during the visit in which she received the Shingrix vaccine.

Eleven days post-vaccination, the patient presented to her primary care provider’s office with paresthesia spreading bilaterally through the wrists, arms, and lower body with increasing progression for about 3 to 4 days. The patient reported new-onset headaches, fever, and muscle aches in her hips and back. An at-home COVID-19 test was negative, and no upper respiratory symptoms were reported. A history and physical examination performed by her primary care provider found pharyngitis and cephalgia. The working diagnosis was Lyme disease or another tick-borne illness causing meningoencephalitis, which was reported to the patient. The patient reported no known tick or insect bites, though she reported it was difficult to be sure. General lab work and a Lyme/tick panel were ordered. The patient was prescribed doxycycline to be taken twice daily for 21 days. Her Lyme and tick panels both came back negative.

Within hours of her primary care office visit, the patient presented to the emergency department with fevers, numbness, tingling in her body, and a shuffling, stiff gait. A lumbar puncture showed a white cell count of 452 (as shown in Table 1), subsequently leading to treatment for suspected meningitis with meropenem, dexamethasone, and vancomycin. The patient was then admitted to the hospital 12 days post-vaccination due to her rapidly progressive neurological symptoms.

Upon admission, her MRI results were as follows: unremarkable for brain pathology, multilevel disc disease with mild spinal canal stenosis of C4–C7, posterior disc bulge at T7–T8 with flattening of the ventral thecal sac and deformity of the ventral spinal cord, and moderate neural foraminal narrowing of L4–S1. The MRI showed no concerns associated with cord signaling in any region. A chest CT with contrast showed that there was no evidence of an intrathoracic malignancy as a contributing factor for her ascending weakness. The EEG showed a normal awake state record, no definite spikes or paroxysms of significance, and no significant epileptiform activities. No electrographic seizures were recorded. No significant focal or lateralizing abnormalities were seen. Even with the patient’s leg shaking, no significant EEG abnormalities were recorded.

Thirteen days post-vaccination, a second MRI without contrast was performed to assess her progressive lower extremity weakness. The MRI found no indications of flow-limiting stenosis or occlusion of proximal intracranial arteries.

An electromyogram of the right upper and bilateral lower extremities came back normal, with the electrodiagnostic evaluation revealing no findings suggestive of cervical radiculopathy, brachial plexopathy, lumbosacral radiculopathy/plexopathy, other peripheral mono/polyneuropathy, or any myopathic process affecting the right upper and bilateral lower extremities.

The working diagnosis for this patient remained broad, but included Lyme disease with atypical presentation, atypical GBS, primary lymphomatous meningitis, CNS leukemia, or a paraneoplastic syndrome. An infectious disease consultation ordered syphilis, HIV, and HSV tests, which all came back negative. Neurology was consulted and started the patient on IVIG and Rocephin. Her neuropathy showed improvement with the initiation of IVIG and Rocephin.

Fourteen days post-vaccination, the patient was entered into evaluation for physical therapy with the goal of ambulating independently on level surfaces, as her neurological symptoms had significantly impacted her mobility. Twice daily physical therapy was performed with the plan to discharge her to a skilled rehabilitation facility when ready. Her physical therapy treatments continued for 16 days.

During this visit, a large calcium oxalate kidney stone weighing 2.3 g was also diagnosed. The patient underwent percutaneous nephrolithotomy with ureteral stent placement 11 days post-vaccination. The kidney stone was reported to be incidental and unlikely to be related to her chief complaint on presentation.

### Outcome and Follow-Up

Upon discharge 17 days post-vaccination, the confirmed diagnosis was never determined for the presenting symptoms of fever, numbness, tingling, and paresthesia, though it was eventually presumed to be a neurological event stemming from the Shingrix vaccination by her providers due to her response to IVIG. The patient reported improvement, albeit minimal, with continuing numbness, tingling, and weakness. Continued improvement was reported to her primary care provider, extending all the way to 74 days post-vaccination, but numbness in her feet and groin persisted, as seen in the complete timeline of symptoms and treatments in Figure 1. At present, the patient reports that her symptoms have nearly (but not completely) subsided, with occasional numbness in her feet and groin. She reports frustration concerning her perceived delay in treatment as well as the lack of an eventual diagnosis. Her case was reported to VAERS in the hope that it can increase patient and provider vigilance.

The patient’s inconclusive diagnosis remains due to incomplete testing as well as the broad differential upon admission that did not include rare adverse events, leaving the patient unsure of how to proceed medically. These lingering questions can persist for patients and providers, leaving all parties involved to speculate based on the circumstances. With continued patient and provider education, we remain hopeful that both parties involved utilize resources, such as VAERS, in order to consider rare, but serious events.

## 3. Discussion

Although a conclusive diagnosis was not reached for this patient, a presumed diagnosis of an adverse neurological event, such as GBS, following vaccine administration was eventually reached. Her symptom onset at 11 days post-vaccination is consistent with the pathophysiology surrounding GBS. She also was taking no other medications and received no other vaccines at the time. Her CSF studies were inconsistent with a typical GBS, but failed to produce any microbial growth indicating infection. In addition, it is less likely that an infection would continue to produce neurologic symptoms as far as 74 days after an event. Lastly, her improvement was seen with the initiation of IVIG, not with the initial treatment regimen that included coverage for meningitis with meropenem and vancomycin. Confounding factors, such as an infection, may have been present, but we believe the likely source of this patient’s prolonged neuropathy to be related to her Shingrix vaccination.

The treatment with (and outcome after using) IVIG suggests a grade 4 serious adverse event (SAE) from the Shingrix vaccine in this patient. We hypothesize that some sort of immune cross-reactivity may have led to her neurological symptoms, although a number of factors could be involved in her symptomatology. The Q2-S1 molecule found in the AS01 adjuvant may be a culprit for the cross-reactivity noted in this case. It has been noted in the literature that an increase in inflammatory cytokines, such as IFN-y, can be observed following the administration of this adjuvant [9]. This is hypothesized to be the driving force behind some of the more common adverse reactions seen, such as fatigue, myalgias, and localized erythema. It is possible that this molecule, or another component of the AS01 adjuvant, is involved in the molecular mimicry that induces neurological symptoms, similar to those reactions found between S. pyogenes and rheumatic fever [10]. These are noted adverse events of the Shingrix vaccine as detailed further in this paper and its surrounding literature.

Progressive neuropathy is a reported adverse event of the Shingrix vaccine. The prescribed course of treatment of neuropathy is multifactorial depending on the etiology of the disease. If an underlying cause such as diabetes is prevalent, the treatment should be focused on this. If a demyelinating neuropathy such as GBS is suspected, treatments include IVIG, corticosteroids, and plasma exchange. The suspicion of a polyneuropathy, such as GBS, is warranted in this patient, as her onset of symptoms 11 days after vaccination and treatment response to IVIG is consistent with other cases.

The Shingrix phase 3 clinical trial demonstrated an increased risk of developing GBS following the administration of the vaccine, with an increased severity in outcomes in adults aged 60 and older [11]. This finding is consistent with case reports of GBS following Shingrix administration. For example, a report described a 76-year-old female who developed GBS 10 days after receiving her first dose of Shingrix. Treatment with intravenous immunoglobulin (IVIG) initially resulted in improvement, but the patient experienced a recurrence of symptoms after discharge. The subsequent treatment with plasma exchange therapy led to a return to baseline [12]. Another case report involved a 79-year-old male who developed GBS 10 days after receiving Shingrix. He was successfully treated with IVIG [13].

In response to these reports and the phase 3 trial findings, the FDA issued a black-box warning for Shingrix in 2021 regarding the potential risk of GBS following vaccination [13]. Previous reports have also documented cases of GBS following Shingrix administration, including a case of an elderly male who developed the rare acute motor sensory axonal neuropathy (AMSAN) variant of GBS. This patient experienced recurrent bouts of rigors, weakness, and gait instability. He achieved a full recovery after plasma exchange but had a recurrent episode 10 months later [13].

While the Shingrix vaccine demonstrated a small increased risk of certain adverse events, like GBS, the morbidity and mortality associated with shingles infections in unvaccinated individuals are far more substantial. Shingles affects approximately 1 million Americans each year, with the lifetime risk being 1 in 3. Complications occur in approximately 13–40% of shingles cases, including debilitating post-herpetic neuralgia, which can persist for a period ranging from months to years. Hospitalizations due to shingles have been estimated at 45,000–75,000 annually. In comparison, based on post-marketing surveillance data, the estimated risk of GBS after Shingrix is only around three excess cases per million doses administered. Most vaccine adverse events, like GBS, are transient and self-limited. In contrast, shingles can lead to severe acute pain during the rash phase, rated by patients as comparable to childbirth or a heart attack. In addition, the post-herpetic neuralgia complication can greatly diminish quality of life in the long term. While any adverse event post-vaccination is concerning, the incidence of most vaccine side effects is quite low. Given the ability of Shingrix to reduce shingles risk by 97% and its ability to mitigate acute and long-term complications in those who still are affected shingles post-vaccination, the benefit–risk profile remains highly favorable for vaccination in the approved age groups. Careful monitoring for adverse events and comparisons to shingles morbidity/mortality remain important, but at present, the benefits appear to considerably outweigh the potential harms.

While pre-licensure clinical trials and short-term post-marketing surveillance provide initial safety data on vaccines, the continued monitoring for adverse events across longer time periods is critical. For example, in the case of Shingrix, the increased risk of GBS was not identified until post-marketing monitoring. Vaccines may have rare risks that emerge or are characterized only after millions of doses are administered. Some adverse events may also manifest at a delayed timescale, months or years after vaccination. As a result, having robust systems for ongoing safety monitoring, like the Vaccine Adverse Event Reporting System (VAERS), is imperative. Healthcare providers play a key role in continually assessing patients’ health post-vaccination and reporting any concerning events, even if unlikely to be related. For Shingrix and other vaccines, obtaining long-term follow-up data would further inform the full safety profile. Rare adverse events need dedicated study. While most vaccines undergo rigorous pre-approval trials, real-world data over many years in large populations reveal the complete picture of both benefits and risks. Continued vigilance and the study of post-licensure adverse events are warranted.

It is important to reiterate that this case report detailing a concerning adverse event following Shingrix vaccination should not be misconstrued as anti-vaccine literature. Documenting and understanding safety signals, even rare ones, allows for proper education and informed benefit–risk analysis. The overwhelming data support the substantial public health benefit of the Shingrix vaccine in appropriate populations. However, transparency regarding full safety profiles is equally important. We advise against making generalizations or overstating implications from limited case reports. Providers should continue recommending this vaccine as per guidelines. Vigilance for risks does not equate to undermining vaccine confidence or efficacy.

In this paper, we reported a probable serious adverse event associated with the Shingrix vaccine. A confirmed diagnosis was never reached, with a working diagnosis that was eventually ruled out. The prevalence of neuropathy associated with Shingrix is low, however not unheard of, as a black box warning for GBS is included on the label. We advocate for the continued awareness of serious neuropathy after Shingrix vaccination to allow for improved patient care and peace of mind, with the goal of educating both patients and providers about this rare event in order to improve awareness, destigmatize efficacious vaccines, and increase surveillance regarding long-term or delayed side effects following vaccinations.

## 4. Conclusions

We advocate for increased awareness and vigilance around rare, but serious adverse events associated with vaccines. We promote the use of VAERS as well as patient–provider communication and education in order to improve outcomes and time to diagnosis. In cases such as this one, we believe there should be an early suspicion of neuropathy with vaccination and appropriate testing, such as CSF studies and EMGs, should be performed swiftly when deemed clinically appropriate. First-line treatments for GBS include IVIG, corticosteroids, and a plasma exchange, all of which should be considered for patients such as this one.

## Figures and Tables

**Figure 1 reports-07-00005-f001:**
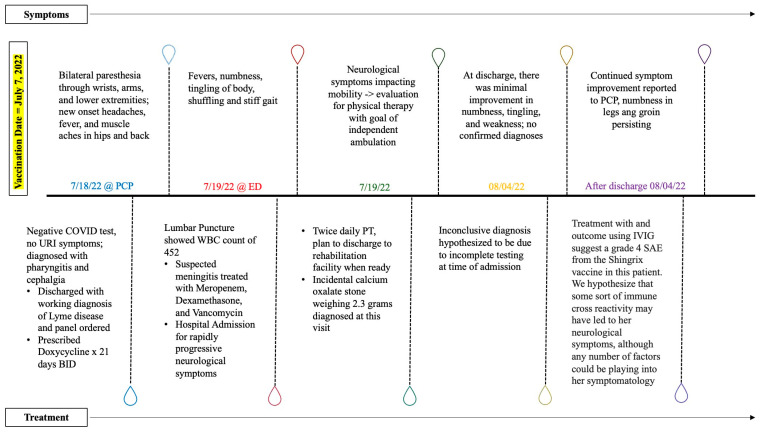
A timeline indicating the symptoms and corresponding treatments.

**Table 1 reports-07-00005-t001:** Pertinent laboratory values upon admission.

Total Protein	8.2 g/dL	High
Glucose	127 mg/dL	High
Alkaline Phosphatase	128 Int_Unit/L	High
Hemoglobin	11.4 g/dL	Low
Hematocrit	33.9%	Low
Red Blood Cell Count	3.55 Mil/uL	Low
Sedimentation Rate	43 mm/h	High
C-Reactive Protein	1.1 mg/dL	High
CSF Glucose	76 mg/dL	High
CSF Protein	90 mg/dL	Critical
CSF RBC Count	19 CMM	High
CSF WBC Count	452 CMM	High
Mean Platelet Volume	12.6	High

## Data Availability

The data presented in this study are available upon request from the corresponding author. The data are not publicly available due to privacy concerns.

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
