# Peer review of "Serious and Progressive Neuropathy Presumably Post-Shingrix Vaccination"

_reports, 2024, doi:10.3390/reports7010005_

Round 1

Reviewer 1 Report

Comments and Suggestions for Authors

The authors presented a new case report which may help to understand and follow up on the post-vaccination adverse reaction. Although there is no clear cut between the "many diagnosis tests harnessed and the results" and their hypothesis that the cause for this case report is the herpes zoster subunit (HZ/su) vaccine.

Although this, the authors requested to present tabulated data for all previous similar case reports due to the herpes zoster subunit (HZ/su) vaccine Shingrix, which helps the reader to compare and follow up.

Reviewer 2 Report

Comments and Suggestions for Authors

The case report by Professor Wons and colleagues is very well done, with accurate description and appropriate and shareable comments. I just have a request for clarification. The patient's medical history lacks any medications or other vaccines taken in the period to exclude pharmacological interactions or other causal or temporal associations.

Author Response

Thank you for pointing this out. We have revised our manuscript to include a mention that she was not taking any other medications and received no other vaccinations at the time of her visit that could have been associated with her symptoms, found on lines 65 and 66. 

"The patient had received no other medications or vaccinations at her visit in which she received the Shingrix vaccine."

Reviewer 3 Report

Comments and Suggestions for Authors

The case reported by Michael Wons et al. "Serious and Progressive Neuropathy Presumably Post Shingrix Vaccination," discusses a severe and progressive neuropathy observed in a patient 11 days after the administration of the Shingrix vaccine, a herpes zoster subunit (HZ/su) vaccine. The patient experienced neuropathy within 11 days of vaccination, which continued to progress until discharge 16 days post-administration. While the occurrence of neuropathy following HZ/su vaccination is exceptionally rare, with an attributable risk of 3 cases per million vaccines, the patient's diagnosis remained inconclusive in this case due to "incomplete testing as well as a broad differential upon admission that may not include rare adverse events, leaving the patient unsure of how to proceed medically."

Despite the low prevalence of neuropathy associated with Shingrix, the report advocates for continued awareness of serious adverse events, emphasizing the need for patient and provider education. The study encourages the use of resources like VAERS (Vaccine Adverse Event Reporting System) to report rare but severe events, but does not clarify that THIS specific case was reported to this system, which is the appropriate outlet for such rare suspected adverse reactions, even when inconclusively linked to vaccination. The report concludes by emphasizing the importance of vigilance regarding long-term or delayed side effects following vaccinations to enhance surveillance and ensure patient safety.

The diagnosis of Guillain-Barré Syndrome (GBS) typically relies on a thorough examination of patient history, neurological assessments, electrophysiological studies, and cerebrospinal fluid (CSF) examinations. However, in this particular case, the results of the conducted tests, including CSF white blood cell count (WBC), fever, and electromyography (EMG), may raise considerations for alternative diagnoses. It is worth noting that the Lyme panel was ordered, but the reported results are not available. Additionally, the EMG results did not align with the characteristics of GBS. The presence of CSF abnormalities and fever in this context may indicate the possibility of another underlying medical condition or process that warrants further investigation.

The mention of 'speculation based on circumstances' in this case report may introduce uncertainty not only for the authors and the patient but ALSO for the reader. This unintended outcome raises concerns about the clarity of the reported case and leaves uncertainty regarding the appropriateness of the medical treatment and course. To enhance the value of the report, it would be beneficial for the authors to provide additional insights or recommendations, offering CLEARER guidance for managing similar cases in the future. This would contribute to a more comprehensive understanding of the presented case and ensure the report aligns with its intended purpose. In doing so, this would help the authors contribute to improved patient care, awareness, and destigmatization of effective vaccines. 

Comments on the Quality of English Language

Minor revisions are needed to improve clarity of presentation.

Round 2

Reviewer 1 Report

Comments and Suggestions for Authors

Thank you,

Author Response

Thank you!

Reviewer 3 Report

Comments and Suggestions for Authors

The concerns about the presentation of the case have been adequately addressed. This is not a novel complication of this vaccine class. Given the ambiguities in presentation and risk that this report raises unnecessary concerns about the safety of such vaccination in the wide public, I am not certain that this case meets criteria based on novelty.

Comments on the Quality of English Language

Minor improvements and polish are still possible.

Author Response

Thank you for your time and input. We will continue to polish the writing.